# BRWD1 orchestrates epigenetic landscape of late B lymphopoiesis

Malay Mandal [1], Mark Maienschein-Cline[2], Patrick Maffucci [3,4], Margaret Veselits[1], Domenick E. Kennedy [1], Kaitlin C. McLean[1], Michael K. Okoreeh[1], Sophiya Karki[5], Charlotte Cunningham-Rundles[3,4] & Marcus R. Clark[1]

Transcription factor (TF) networks determine cell fate in hematopoiesis. However, how TFs cooperate with other regulatory mechanisms to instruct transcription remains poorly understood. Here we show that in small pre-B cells, the lineage restricted epigenetic reader BRWD1 closes early development enhancers and opens the enhancers of late B lympho-poiesis to TF binding. BRWD1 regulates over 7000 genes to repress proliferative and induce differentiation programs. However, BRWD1 does not regulate the expression of TFs required for B lymphopoiesis. Hypogammaglobulinemia patients with *BRWD1* mutations have B-cell transcriptional profiles and enhancer landscapes similar to those observed in *Brwd1*[-/-] mice. These data indicate that, in both mice and humans, BRWD1 is a master orchestrator of enhancer accessibility that cooperates with TF networks to drive late B-cell development.

[1] Department of Medicine, Section of Rheumatology and Gwen Knapp Center for Lupus and Immunology Research, University of Chicago, Chicago, Illinois, USA. [2] Core for Research Informatics, University of Illinois at Chicago, Chicago, Illinois, USA. [3] Immunology Institute, Icahn School of Medicine at Mount Sinai, New York, NY, USA. [4] Division of Clinical Immunology, Department of Medicine, Icahn School of Medicine at Mount Sinai, New York, NY, USA. [5] Department of Research Biology, Genentech, South San Francisco, California, USA. Correspondence and requests for materials should be addressed to M.M. (email: mmandal@medicine.bsd.uchicago.edu) or to M.R.C. (email: mclark@uchicago.edu)

B-cell development consists of sequential and mutually exclusive states of proliferation and immunoglobulin gene recombination[1,2]. Following in-frame *Igμ* recombination, the expressed immunoglobulin μ-chain assembles with surrogate light chain (λ5 and VpreB), Igα, and Igβ to form the pre-B cell receptor complex (pre-BCR). The pre-BCR, in concert with cell extrinsic (IL-7R)[3] and intrinsic[4] cues direct large pre-B-cell proliferation, followed by cell cycle exit and *Igk* recombination in small pre-B cells[1]. Aberrations in the mechanisms that segregate proliferation from *Igk* recombination can lead to either immunodeficiency or genomic instability and leukemic transformation[5,6].

Many of the signaling mechanisms that coordinate proliferation and *Igk* recombination in pre-B cells have recently been elucidated. Downstream of the pre-BCR, E2A, and the interferon-regulatory factor family (IRF) members IRF4 and IRF8, direct cell cycle exit and open the *Igk* locus for recombination[3,7–10]. *Igk* recombination also requires escape from IL-7R signaling, which results in loss of STAT5 activation, downregulation of cyclin D3, and derepression of the *Igk* locus[11,12]. Coordinate loss of IL-7R-dependent PI-3K activation derepresses FOXO1 and FOXO3, which then induces *RAG1* and *RAG2*[10,13,14]. Feedback inhibition is provided by RAG-mediated dsDNA breaks which activate NF-κB2 that then represses expression of pre-BCR signaling intermediates[4]. Through these and other feed-forward and backward loops[1], transcription factor (TF) networks enforce context-appropriate cell-fate decisions.

The primary targets of these TFs are enhancers, which, following TF binding, productively interact with cognate gene promoters to regulate specific gene expression programs[15]. Indeed, differential activation of enhancers appears to determine most cell type-specific gene expression[16]. Regulation of enhancer accessibility to TFs is an area of intense interest. Some TFs, such as PU.1, IKZF1 (Ikaros), and EBF1, can invade enhancers occupied by nucleosomes and, either singly or in combination, open the enhancers to subsequent TFs[17–19]. In some cases, these pioneering TFs open enhancers by recruiting histone methyltransferases and acetyltransferases[19].

Active enhancers are typically marked by histone H3 lysine 27 acetylation (H3K27ac) and histone H3 lysine 4 monomethylation (H3K4me1), whereas silent enhancers are usually enriched for histone H3 lysine 27 trimethylation (H3K27me3)[20,21]. These post-translational modifications form recognition motifs for epigenetic reader complexes[22,23] containing chromatin remodelers such as SWI/SNF (switching defective/sucrose non-fermenting)[24]. However, not all enhancers are opened by pioneer TFs[19]. Furthermore, it is unclear how molecular mechanisms downstream of TFs and histone methyltransferases and acetyltransferases are regulated and how these regulatory mechanisms determine enhancer landscapes.

Recent studies of B lymphopoiesis have identified critical TFs and some of their targets[1,8,9,11–13,25]. However, remarkably little is known regarding the epigenetic regulation of B lymphopoiesis. Some TFs important for B lymphopoiesis, including E2A, Pax, and STAT5, can modulate enhancer accessibility through the recruitment of histone modifying and remodeling complexes[11,26,27]. Furthermore, EZH2, which generates H3K27me3, plays important roles in both repressing *Igk* accessibility[11] and regulating p53[28]. Recruitment of the RAG proteins, and assembly of the recombination center[2], requires H3K4me3. Downstream of histone post-translational modifications, mutations in humans have implicated epigenetic readers such as the bromodomain and extraterminal (BET) domain protein BRD4 in peripheral leukemias[29,30]. However, the role of epigenetic readers in normal B lymphopoiesis is poorly understood.

We have previously demonstrated that the BROMO and WD40 domain containing epigenetic reader BRWD1 is necessary for opening the Jκ genes, assembly of the RAG recombination center, and subsequent *Igk* recombination[31]. The expression of BRWD1 is lineage and stage specific and thereby contributes to restricting *Igk* recombination to the small pre-B-cell stage. However, BRWD1 binds to numerous sites across the genome[31], suggesting that it could play additional roles in B lymphopoiesis.

Here we demonstrate that BRWD1 orchestrates a genome-wide reordering of enhancer accessibility by both silencing early developmental enhancers and opening those critical for late B lymphopoiesis to TF binding. Additionally, BRWD1 inhibits proliferation by coordinately repressing *Myc* and MYC's downstream targets. Interestingly, *BRWD1* mutations are relatively common in patients with idiopathaic hypogammaglobulinemia. Furthermore, analyses of cells from patients with *BRWD1* mutations reveals a similar transcriptional and epigenetic program as that observed in *Brwd1*[-/-] mice including the activation of *MYC* and MYC-dependent pathways. Overall, this study identifies a previously unrecognized mechanism, in both mice and humans, for remodeling the enhancer landscape of late B lymphopoiesis.

## Results

**BRWD1 orchestrates transcription of late B-cell development.**
RNA-Seq (Supplementary Table 1) of *Brwd1*[-/-] and wild-type (WT) small pre-B cells indicated that over 7000 genes were dysregulated ($q < 0.05$) in *Brwd1*[-/-] small pre-B cells (Fig. 1a and Supplemental data 1), with enhanced expression (3901 genes) being more common than the decreased expression (3405). For example, BRWD1 repressed *Ccr9* (Fig. 1b), and CCR9 surface densities were intermediate between pro- and small pre-B cells (Fig. 1c). A similar expression pattern was observed for Flt3 (Fig. 1d, e). In contrast, normal upregulation of the IL-2 receptor (*Il2ra*, CD25) (Fig. 1f, g) and CXCR4 (Fig. 1h, i) in pre-B cells was diminished in *Brwd1*[-/-] cells, with surface expression levels intermediate between WT pro- and small pre-B cells. These examples suggest that BRWD1 both represses early, and induces late, developmental genes.

To test this, we grouped all differentially expressed genes during B lymphopoiesis (one-way ANOVA, $q < 0.05$, 13387 genes, Immgen[32]) by their expression, before, at, and after the small pre-B-cell stage (Fig. 1j). Remarkably, most genes were clustered into two large expression groups. The first cluster (Cluster 1) contained genes that were highly expressed in early B-cell development that were then downregulated starting in small pre-B cells. The second cluster (Cluster 2) contained genes that were induced after the small pre-B-cell stage. These data indicate that the small pre-B-cell stage is a watershed between early and late B-cell development, in which there is a global reordering of transcriptional programs.

BRWD1 preferentially repressed Cluster 1 genes ($p = 4.1 \times 10^{-441}$) and upregulated Cluster 2 ($p = 4.9 \times 10^{-423}$) (Fig. 1k). At the small pre-B-cell stage, cells stop proliferating and initiate *Igk* recombination[1]. Indeed, ontology analysis demonstrated that BRWD1 induced B-cell activation and differentiation transcription programs, while repressing genes involved in proliferation and metabolism (Supplementary Fig. 1a, b and Supplementary Table 2). These data indicate that BRWD1 controls the transition between early B-cell proliferative and late differentiative developmental programs.

**BRWD1 regulates chromatin accessibility.** We next used ATAC-Seq to examine how BRWD1 regulated chromatin accessibility (Fig. 2 and Supplementary Table 3). In WT cells, progression from the pre-pro B to small pre-B-cell stage was associated with progressive net loss of chromatin accessibility (Fig. 2a). At each stage, new accessibility sites appeared, but these

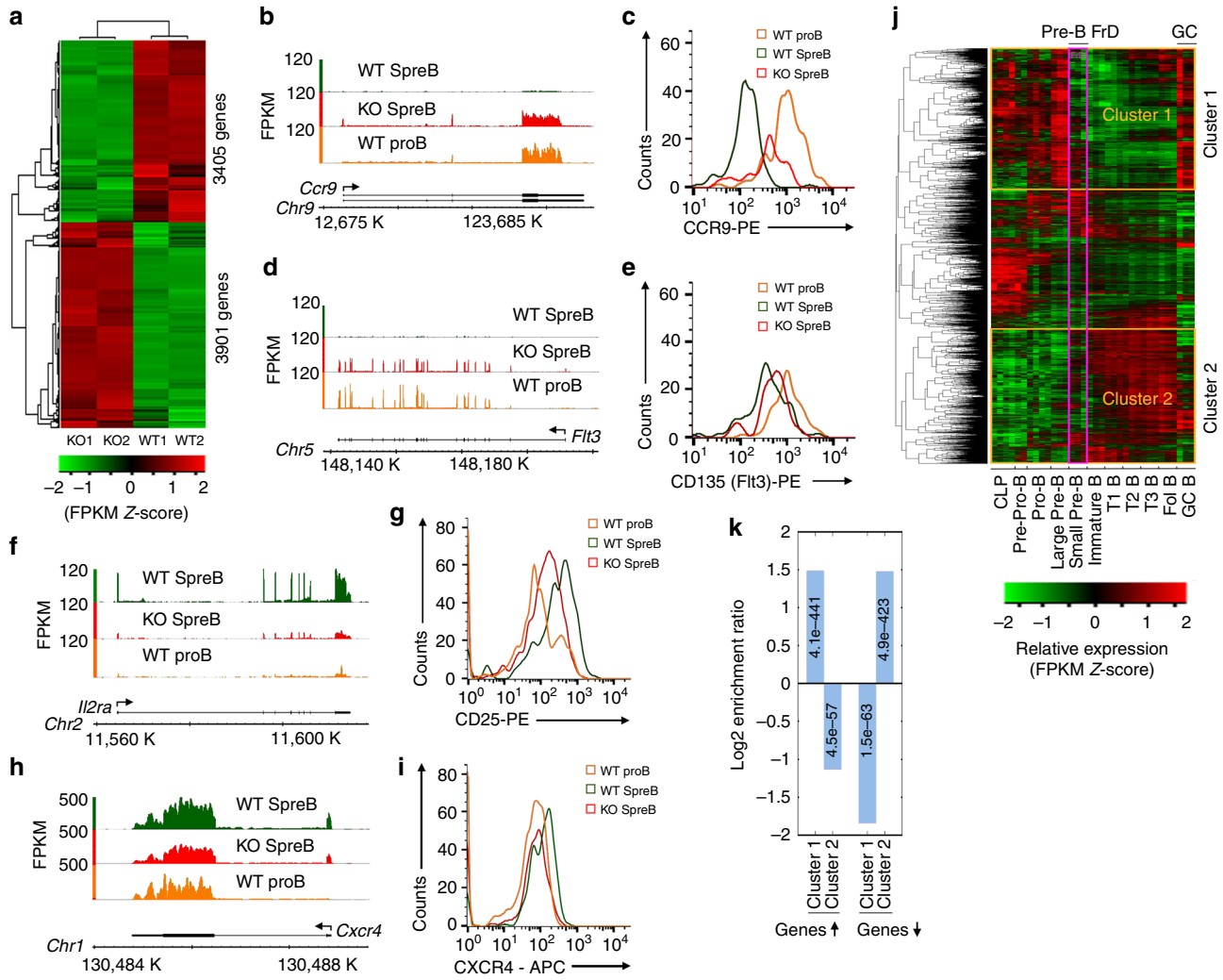

**Fig. 1** BRWD1 orchestrates transcriptional programs of late B-cell development. **a** Heatmap of RNA-Seq results with clustering of upregulated and downregulated genes in *Brwd1*[-/-] vs. WT small pre-B cells (*q* < 0.05, replicates shown). **b**, **e** Representative RNA-Seq alignments for *Ccr9* (**b**) and *Flt3* (**d**) in WT and *Brwd1*[-/-] small pre-B cells. Corresponding surface expression of CCR9 (**c**) and FLT3 (**e**) on indicated cell populations as assessed by flow cytometry. **f**, **i**. Representative RNA-Seq alignments for *Il2ra* (**f**) and *Cxcr4* (**h**) in WT and *Brwd1*[-/-] small pre-B cells. Corresponding surface expression of CD25 (**g**) and CXCR4 (**i**) on indicated cell populations as determined by flow cytometry. **j** Gene expression microarray data from ImmGen (*10*) covering ten B-cell development and maturation stages including common lymphoid progenitor (CLP), and transitional (T1-3) and germinal center (GC). A total of 13387 differentially expressed genes (one-way ANOVA, q < 0.05) were hierarchically clustered. **k** Log₂ enrichment of Cluster 1 and Cluster 2 genes with lower and higher transcript abundance in *Brwd1*[-/-] small pre-B cells than in WT small pre-B cells. Corresponding *P*-values (unpaired *t* test) indicated

represented a minority of all changed sites. For example, in transition from large to small pre-B cells, total accessible sites decreased from 63,492 to 36,276 including 3694 new open sites. Upon transit to the immature B-cell stage (Fig. 2b), this trend reversed and there was a large net gain in accessibility including 29,314 sites not observed in immediate precursor populations.

Remarkably, there were 82,295 accessible sites in *Brwd1*[-/-] small pre-B cells (Fig. 2c). Comparison to other developmental stages indicated that many of these new accessibility peaks could also be found in WT pre-pro-B (35,828), pro-B (26,127), and large pre-B (17,713 peaks) cells. Conversely, 2,943 sites failed to open in *Brwd1*[-/-] small pre-B cells.

Many accessibility peaks in *Brwd1*[-/-] small pre-B cells were also present in WT immature B cells (Fig. 2d). However, most of these peaks were also found in WT pre-pro B cells. Furthermore, de novo accessibility peaks observed in WT immature B cells were not regulated by BRWD1. Overall, these data suggest that BRWD1 broadly represses chromatin associated with early B-cell

developmental stages while opening more restricted chromatin regions associated with the small pre-B-cell stage.

We next examined the relationships between BRWD1-dependent changes in transcription and chromatin accessibility. Provided in Fig. 2e are quantile (Q–Q) plots of RNA differential statistics (*Brwd1*[-/-] vs. WT) as a function of increased (up), decreased (down), or no change (noc) in chromatin accessibility in *Brwd1*[-/-] small pre-B cells. The relationships between chromatin accessibility and transcription were then considered as a function of distance from transcription start sites (TSSs). Increased accessibility in *Brwd1*[-/-] small pre-B cells was associated with increased transcription up to 500 kb removed from TSSs. In contrast, decreased accessibility was clearly associated with diminished transcription within 5 kb of TSSs. However, this effect was largely lost beyond 5 kb. This data suggest that BRWD1 represses transcription through both long- and short-range mechanisms from TSSs, while it primarily enhances transcription through mechanisms proximate to TSSs.

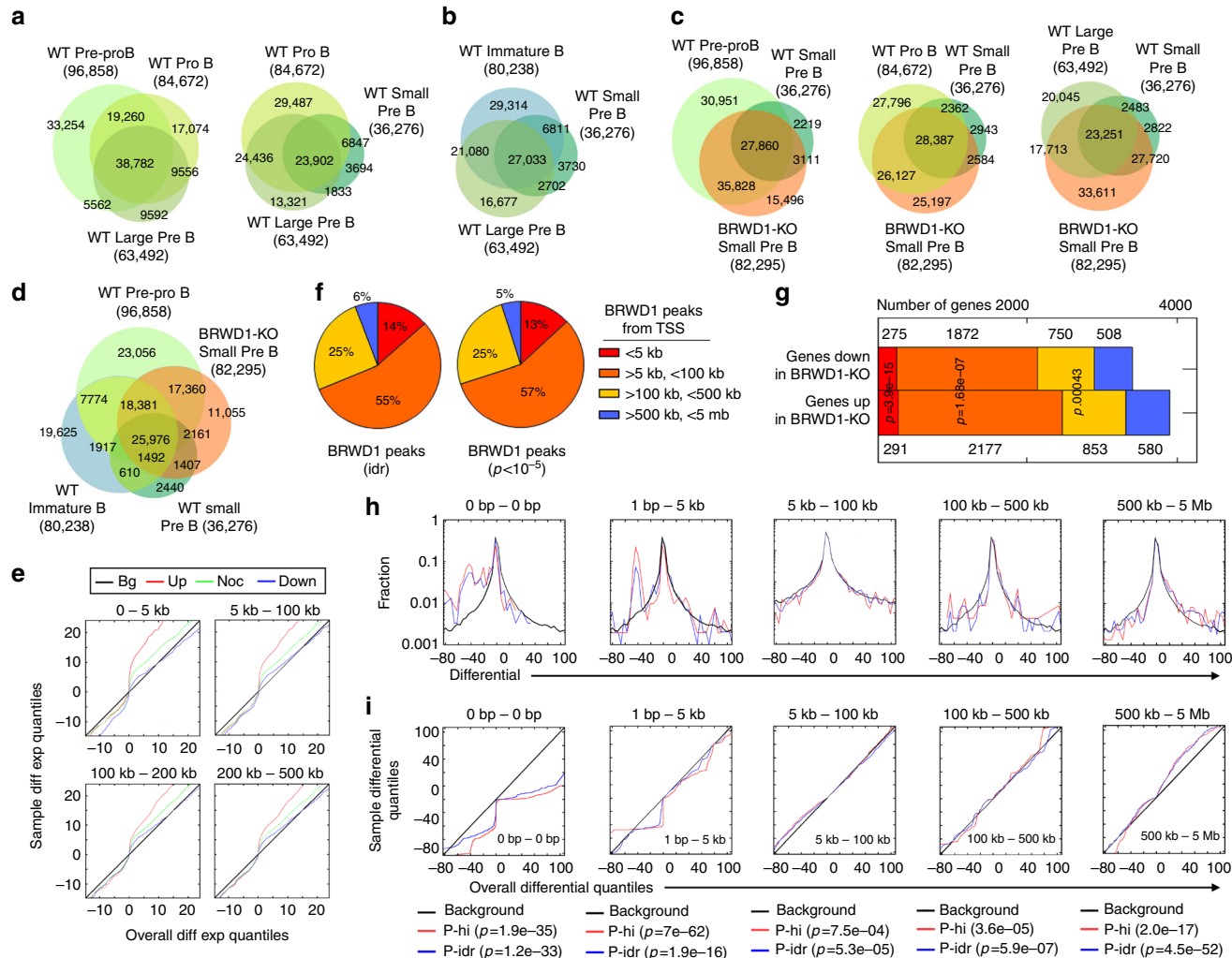

**Fig. 2** BRWD1 regulates chromatin accessibility in late B-cell development. **a, b** Total and overlapping open chromatin peaks (ATAC-Seq) in indicated flow-purified WT B-cell progenitor populations. Total number of peaks for each population shown in parentheses with number in each Venn region indicated. **c** Total and overlapping open chromatin peaks in indicated flow-purified *Brwd1*$^{-/-}$ B-cell progenitor populations. **d** Total and overlapping open chromatin peaks in indicated flow-purified B-cell progenitor populations. **e** Q-Q plots demonstrating relationship between mRNA expression (RNA-Seq) and accessibility (ATAC-Seq) in WT and *Brwd1*$^{-/-}$ small pre-B cells. Quantiles of overall background distribution (Bg, black) and quantiles for accessibility associated with increased (up, red), decreased (down, blue), or unchanged (noc, green) open chromatin (*Brwd1*$^{-/-}$ vs. WT) against transcription (*Brwd1*$^{-/-}$ vs. WT) as a function of distance from transcription start sites (0–5 kb, 5–100 kb, 100–200 kb, and 200–500 kb). **f** Distribution of BRWD1 ChIP-seq peaks called with different statistical parameters by distance from Transcriptional start site (TSS) as described in the figure. **g** Comparison of BRWD1 ChIP-Seq peaks with RNA-Seq data from WT and *Brwd1*$^{-/-}$ small pre-B cells. Genes downregulated or upregulated in *Brwd1*$^{-/-}$ cells were grouped by distance from BRWD1 ChIP-Seq peaks (idr, < 5 kb, 5 kb < peak < 100 kb, 100 kb < peak < 200 kb, and 200 kb < peak < 5 Mb). Statistics refer to significance of enrichment for differentially regulated genes at given distance from a BRWD1 peak. **h, i** Differential histograms **h** and Q-Q plots (**i**) to demonstrate the effect of distances (0–5 Mb) of BRWD1 binding (ChIP-Seq) to accessibility (ATAC-Seq). Q–Q plots described the quantiles of the background distribution of accessibility (overall distribution) against the quantiles for BRWD1 ChIP-Seq peaks. P-hi, BRWD1 ChIP-Seq peaks called with $P < 10^{-7}$ over small pre-B input; P-idr, BRWD1 ChIP-Seq peaks called using Irreproducible Discovery Rate (IDR)

To determine the relationship between BRWD1 binding and gene expression (RNA-Seq), we first analyzed BRWD1 ChIP-Seq[31] peaks using two calling methods, irreproducible discovery rate (idr, $q < 10^{-6}$), or reproducible peaks with a cut-off of $p < 10^{-5}$ significance. The idr was more rigorous providing 1843 peaks, while $p < 10^{-5}$ identified 9474 BRWD1 chromatin-bound sites. We then compared the relationship between BRWD1-bound sites and distance to TSSs. Remarkbly, irrespective of peak calling method, the distance relationship between BRWD1 binding and distance to TSSs was remarkably similar (Fig. 2f). These data suggest that the method of peak calling does not affect the apparent relationship between where BRWD1 is bound and the

distance to potential gene targets. For subsequent analysis, we used idr called ChIP-Seq peaks.

We next examined the distance relationships between BRWD1-bound sites and genes differentially regulated by BRWD1. Interestingly, the TSSs of only 566 differentially regulated genes (7.7%) were within 5 kb of a BRWD1-bound site (Fig. 2g). Even at distances of up to 500 kb from a BRWD1-bound site, only 4615 (63.2%) of differentially expressed genes were captured. Distances of up to 5 mb from BRWD1-bound sites captured essentially all differentially expressed genes. A similar pattern was seen when examining ΔATAC-Seq peaks. However, the effect on accessibility changed as a function of distance from

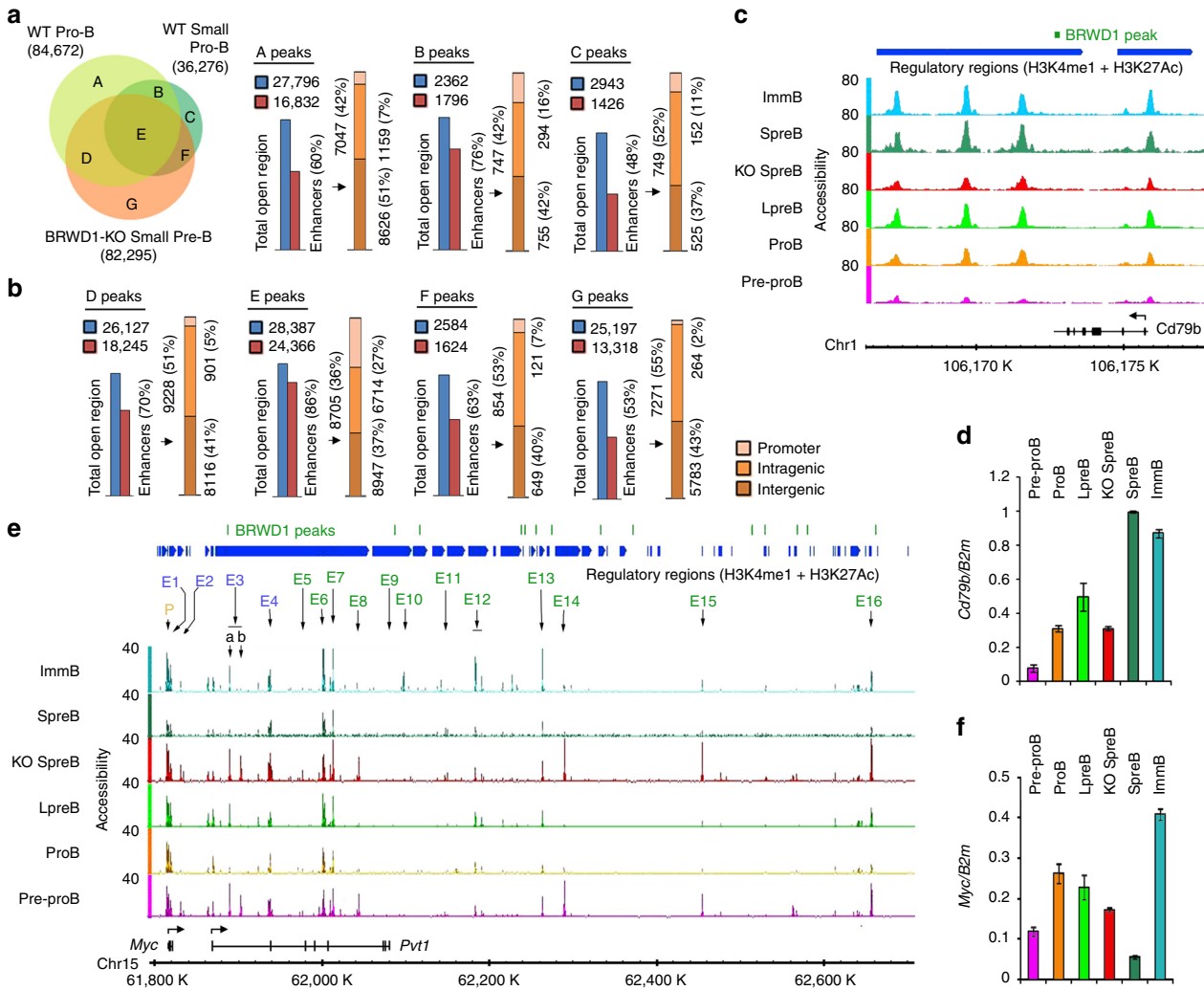

**Fig. 3** BRWD1 regulates transcription by remodeling the enhancer accessibility. **a** Total and overlapping chromatin accessible regions in indicated B-cell progenitors with each unique area in the Venn diagram designated with a letter. **b** Percentage of open chromatin regions identified as enhancers (H3K4me3+H3K27ac+) in different chromatin regions (**a**). The number and percentage of enhancers of each open chromatin region present in different genomic regions. Promoter (-1kb to +500 bp from TSS), Intragenic, or Intergenic. **c**, **d** Accessibility (open chromatin, ATAC-Seq) profile (**c**) and corresponding quantitative mRNA expression (**d**) of *Cd79b* (encoding Igβ) throughout B-cell development stages from WT pre-pro-B to immature B cells and in *Brwd1*[-/-] small pre-B cells. BRWD1 binding sites and enhancer regions (H3K4me3+H3K27ac+) were shown. **e** Accessibility profile of *Myc* during B lymphopoiesis and in *Brwd1*[-/-] small pre-B cells. **f** Quantitative mRNA expression of *Myc* in indicated WT progenitors and *Brwd1*[-/-] small pre-B cells. Data are from one experimental representative of $n = 2$ (**c**,**e**) independent experiments or are presented as average ± s.d. ($n = 3$) (**d**,**f**)

BRWD1-bound sites. Shown in Fig. 2h (Supplementary Table 4) is the differential probability distribution of accessibility peaks in *Brwd1*[-/-] vs. WT small pre-B cells as a function of distance from BRWD1-bound sites in WT small pre-B cells. At sites of BRWD1 binding (0 bp), there is an over-representation of diminished accessibility peaks in *Brwd1*[-/-] cells which persisted to about 5 kb. However, at distances beyond 100 kb, there was an over-representation of open peaks in *Brwd1*[-/-] small pre-B cells that persisted up to 5 mb. The same relationships between distance from BRWD1-bound sites and BRWD1-dependent modulation of chromatin accessibility was evident when the data were displayed as Q–Q plots (Fig. 2i). These data indicate that BRWD1 enhances the accessibility and transcription locally, but can repress both at distances up to 5 mb away.

**BRWD1 acts at enhancers**. Comparison of ATAC-Seq with relevant H3K27Ac and H3K4me1 ChIP-Seq data sets[26,33] revealed that BRWD1 preferentially regulated the enhancers

(Fig. 3a, b). Each unique area in the accessibility Venn diagram for WT Pro-B, WT small pre-B, and *Brwd1*[-/-] small pre-B cells was given a letter (Fig. 3a), which corresponded to bar graphs (Fig. 3b) demonstrating percent of peaks at enhancers and the number of these enhancer peaks that were within 5 kb of promoters or were within intragenic or intergenic regions. Enhancers repressed by BRWD1 (areas 'D' and 'G') were primarily intragenic or intergenic, with only 5% and 2% lying near the TSSs. In contrast, 11% of BRWD1-dependent small pre-B cell accessibility peaks ('C') were near the promoters. This is consistent with BRWD1 preferentially increasing the accessibility at enhancers within 5 kb of TSSs (Fig. 2e, g). Examples of this included *Cd79b* (encoding Igβ, Fig. 3c, d) and *Cxcr4* (Supplementary Fig. 2a, b), in which increased expression was associated with opening of a TSS proximate enhancer.

Examples of BRWD1-mediated enhancer silencing included *Mpeg1*, *Dtx4*, *Fam111a*, *Lef1*, *Ccr9* (Supplementary Fig. 2c–j), and *Myc* (Fig. 3e). *Myc* has 14 known enhancers including four

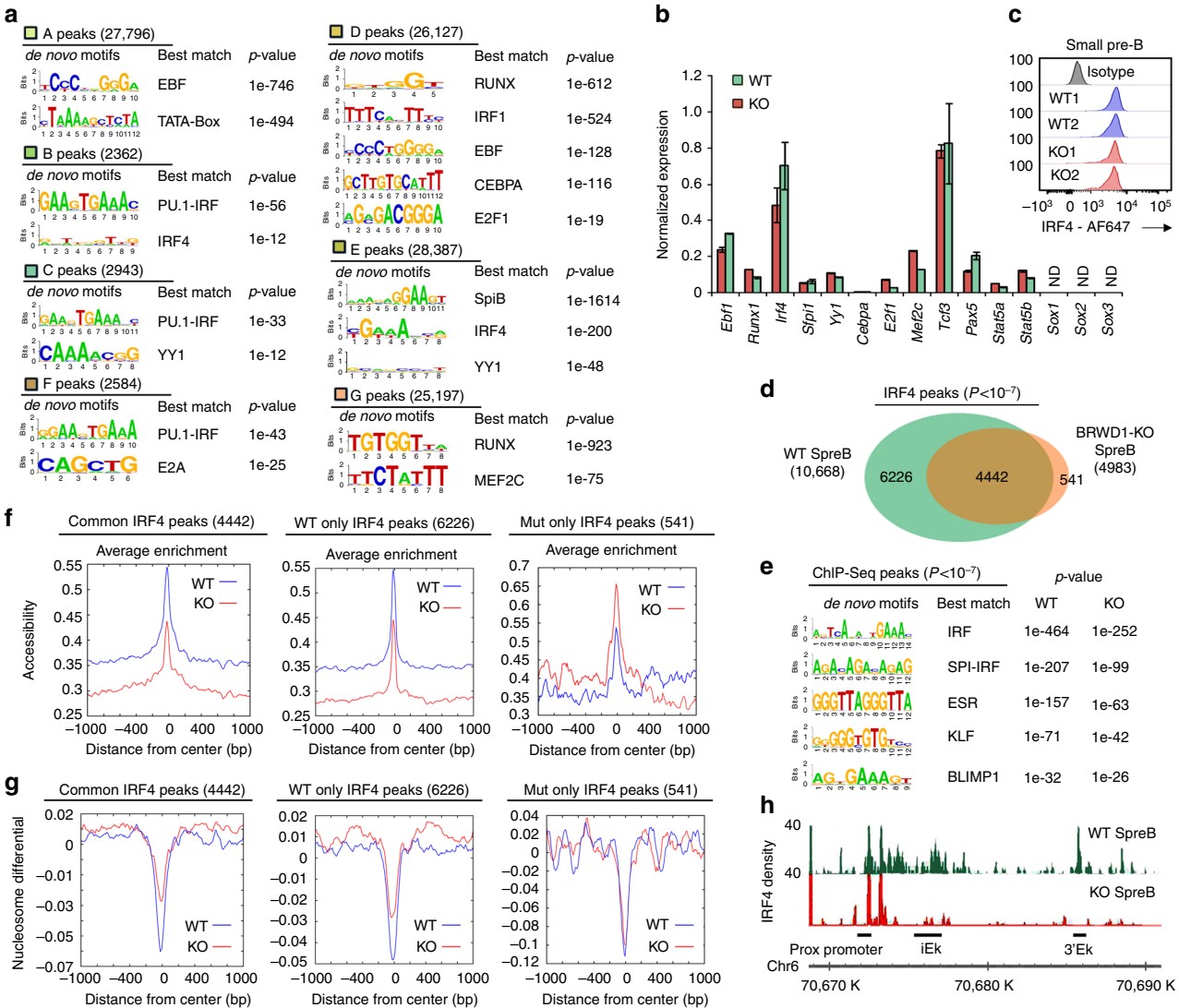

**Fig. 4** BRWD1 determines the genomic targets accessible to TF critical for B-cell fate. **a** De novo DNA sequence motifs identified in different open chromatins regions described in Fig. 3a. **b** Quantitative mRNA expression of indicated transcription factors in purified WT and *Brwd1*[-/-] small pre-B cells. **c** Flow cytometry analysis of intracellular total IRF4 in small pre-B cells isolated from WT and *Brwd1*[-/-] bone marrow; gray-isotype-matched control antibody. **d** Overlap of IRF4 ChIP-Seq peaks ($P < 10^{-7}$) from purified WT and *Brwd1*[-/-] small pre-B cells. **e** De novo DNA sequence motifs identified in IRF4 ChIP-Seq peaks in WT and *Brwd1*[-/-] small pre-B cells. **f, g** DNA footprint analyses of ±1 kb region for WT and *Brwd1*[-/-] small pre-B cells at common IRF4 peaks, WT-only IRF4 peaks and *Brwd1*[-/-] only IRF4 peaks centered on 0 for accessibility (**f**) and nucleosome positions (**g**). **h** ChIP-Seq profile of IRF4 at the intronic kappa enhancer (iEk) and 3′ kappa enhancer (3′Ek) of immunoglobulin kappa (*Igκ*) light chain locus in purified WT and *Brwd1*[-/-] small pre-B cells. Data are from combination (**d**) of independent experiments (*n* = 2) or are presented as average ± s.d. (*n* = 2) (**b**)

(E1–E4) active in embryonic stem cells and an additional 10 enhancers that are accessible in the peripheral activated B cells[16]. In pre-pro B cells, many of these 14 enhancers were accessible with an additional three putative novel enhancers, E3b, E15, and E16, readily apparent (Fig. 3e). With progression to the small pre-B-cell stage, many enhancers including E3, E8, E14, E15, and E16 were silenced and this was associated with binding of BRWD1 at several sites throughout the locus. These early developmental enhancers became accessible in *Brwd1*[-/-] small pre-B cells such that the *Myc* enhancer landscape resembled that observed in pre-pro B cells (Fig. 3e). Immature B-cell enhancers E10, E12, and E13 remained inaccessible in *Brwd1*[-/-] cells, indicating that BRWD1 specifically silenced early developmental enhancers. In addition to repressing *Myc* expression (Fig. 3f), BRWD1 also repressed the enhancers of Myc genomic targets[34] (Supplementary Fig. 3a, b). This was associated with enhanced expression of Myc and its effectors *Tsr1* and *Gar1* (Supplementary Fig. 3c–f)

and general upregulation of Myc-dependent metabolic pathways in *Brwd1*[-/-] small pre-B cells (Supplementary Fig. 3g).

**BRWD1 determines genomic targets accessible to TFs**. We next examined if those enhancers activated and repressed by BRWD1 contained common DNA motifs. As shown in Fig. 4a, the pro-B-cell accessibility peaks repressed by BRWD1 ('D' peaks) were enriched in motifs for TFs involved in early B-cell development (EBF) and hematopoietic development (RUNX, CEBPA). In contrast, BRWD1-dependent small pre-B accessibility peaks ('B' and 'C' peaks) were enriched for ETS (e.g., PU.1), IRF4, and YY1 binding sites, which are TFs (Fig. 4a) required for late B-cell development[9,35].

While BRWD1 modulated the accessibility of TF binding sites, expression of TF genes critical for lymphopoiesis and myelopoiesis[9,11,26,36–38], including *Ebf1*, *Runx1*, *Irf4*, *Sfpi1*

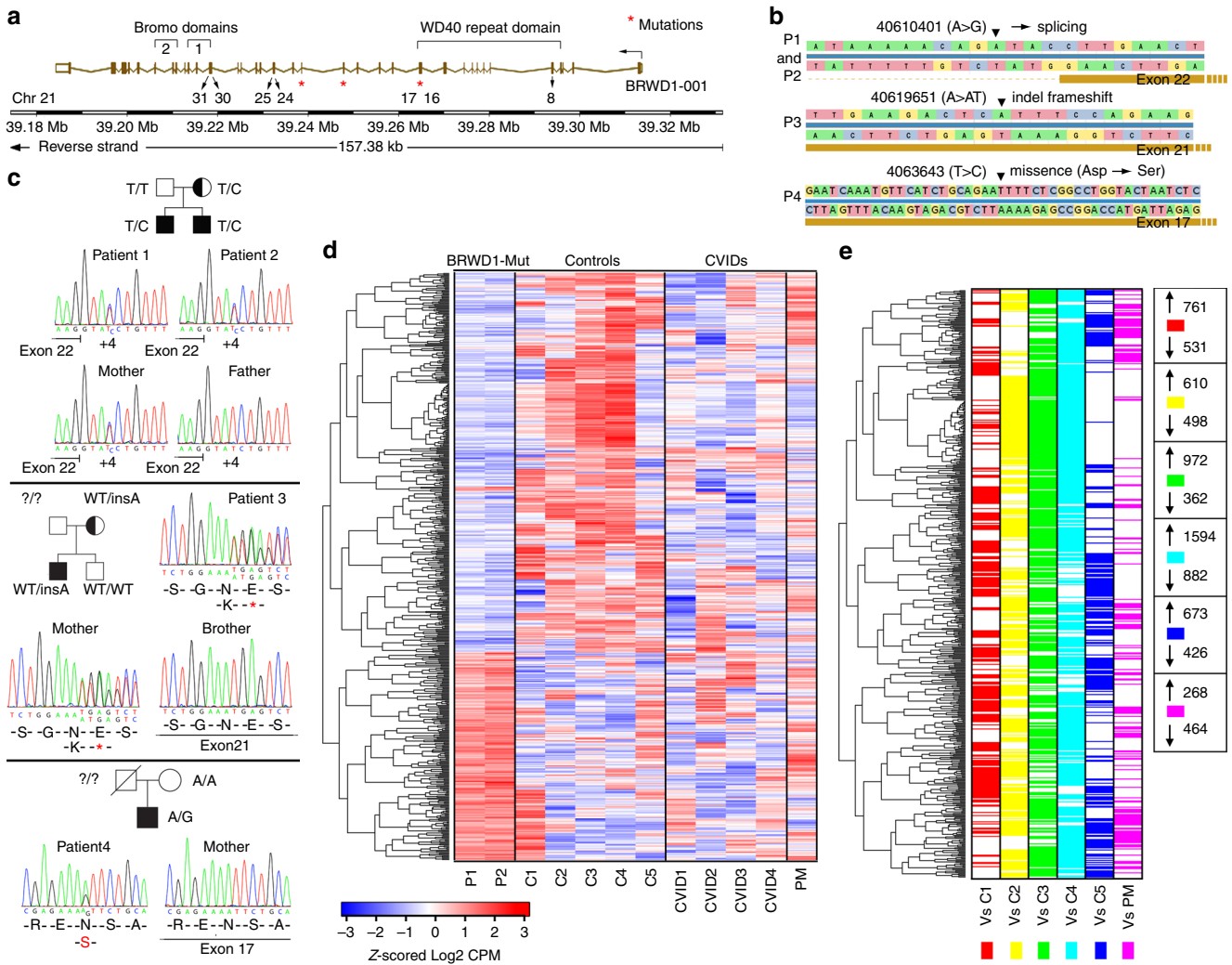

**Fig. 5** *BRWD1* mutations in human hypogammaglobulinemia. **a** Human *BRWD1* with patient mutations (*) and exons encoding WD40 and bromo domains noted. **b** Details of the mutations in *BRWD1* locus of hypogammaglobulinemic patients. **c** Sanger sequencing confirming the mutations in *BRWD1* locus in chromosome 21 of idiopathic hypogammaglobulinemia patients compared to their healthy relatives. **d** Hierarchical clustering of differentially expressed genes identified at $q < 0.05$. EBV-transformed cell lines from controls (C), CVID patients, patients (P) or patients' mother (PM) shown. **e** Pairwise comparison of gene expression in patients vs. indicated controls or PM. The number of genes up (↑) and downregulated (↓) in each comparison provided

(PU.1), *Yy1*, *Cebpa*, *E2f1*, *Mef2c*, *Tcf3e2a* (E2A), *Pax5*, *Stat5a-b*, and *Sox1-3* were similar in *Brwd1*⁻/⁻ and WT small pre-B cells (Fig. 4b). For IRF4, normal mRNA levels were associated with normal protein expression as assessed by intracellular flow cytometry of *Brwd1*⁻/⁻ and WT small pre-B cells (Fig. 4c). These data suggest that BRWD1 determines the genomic targets accessible to TFs critical for B-cell fate decisions without changing the expressions of TFs.

To examine this possibility, we performed IRF4 ChIP-Seq from WT and *Brwd1*⁻/⁻ small pre-B cells. We chose IRF4, as it cannot bind DNA occupied by nucleosomes[17]. IRF4 bound 10,668 sites in WT, but only 4983 sites in *Brwd1*⁻/⁻ cells (Fig. 4d). While the sites were generally lost in *Brwd1*⁻/⁻ cells, 541 new IRF4 binding sites were apparent. In all peak subsets, IRF4 binding sites were enriched (Fig. 4e). Furthermore, in all peak subsets, IRF4 preferentially bound to the open and nucleosome-free sites (Fig. 4f, g). Examination of IRF4 sites in the *Igk* intronic (iEk) and 3′Ek enhancers of *Igk* locus[7,9] revealed dramatically reduced IRF4 binding in *Brwd1*⁻/⁻ small pre-B cells (Fig. 4h). In contrast, IRF4 binding at the proximal *Jk* promoter was similar in WT and *Brwd1*⁻/⁻ small pre-B cells. These data demonstrate that, independent of TF expression, BRWD1 coordinately regulates

the enhancer accessibility genome-wide to induce the transcription programs of B late lymphopoiesis and represses those of early development.

**BRWD1 mutations in human hypogammaglobulinemia.** To understand the importance of BRWD1 in human immunity, we examined 50 patients with hypogammaglobulinemia[39] who had undergone whole-exome sequencing and for which no known mutations associated with B-cell deficiency had been found. Remarkably, four of these patients (clinical characteristics in Supplementary Table 5) had mutations in the *BRWD1* gene. The distribution of these mutations is provided in Fig. 5a, b and family trees in Fig. 5c. Two patients (P1 and P2) were brothers and had a mutation predicted to affect splicing 3′ of exon 22. Patient 3 had an indel mutation in exon 21 that led to a frameshift and premature stop codon, while P4 had a missense mutation in exon 17 (encoding the WD40 repeat domain) predicted to change an asparagine to a serine. Patient 4 had no detectable peripheral B cells or serum immunoglobulins while the three other patients had severely diminished, but detectable B cells and immunoglobulin levels (Supplementary Table 5).

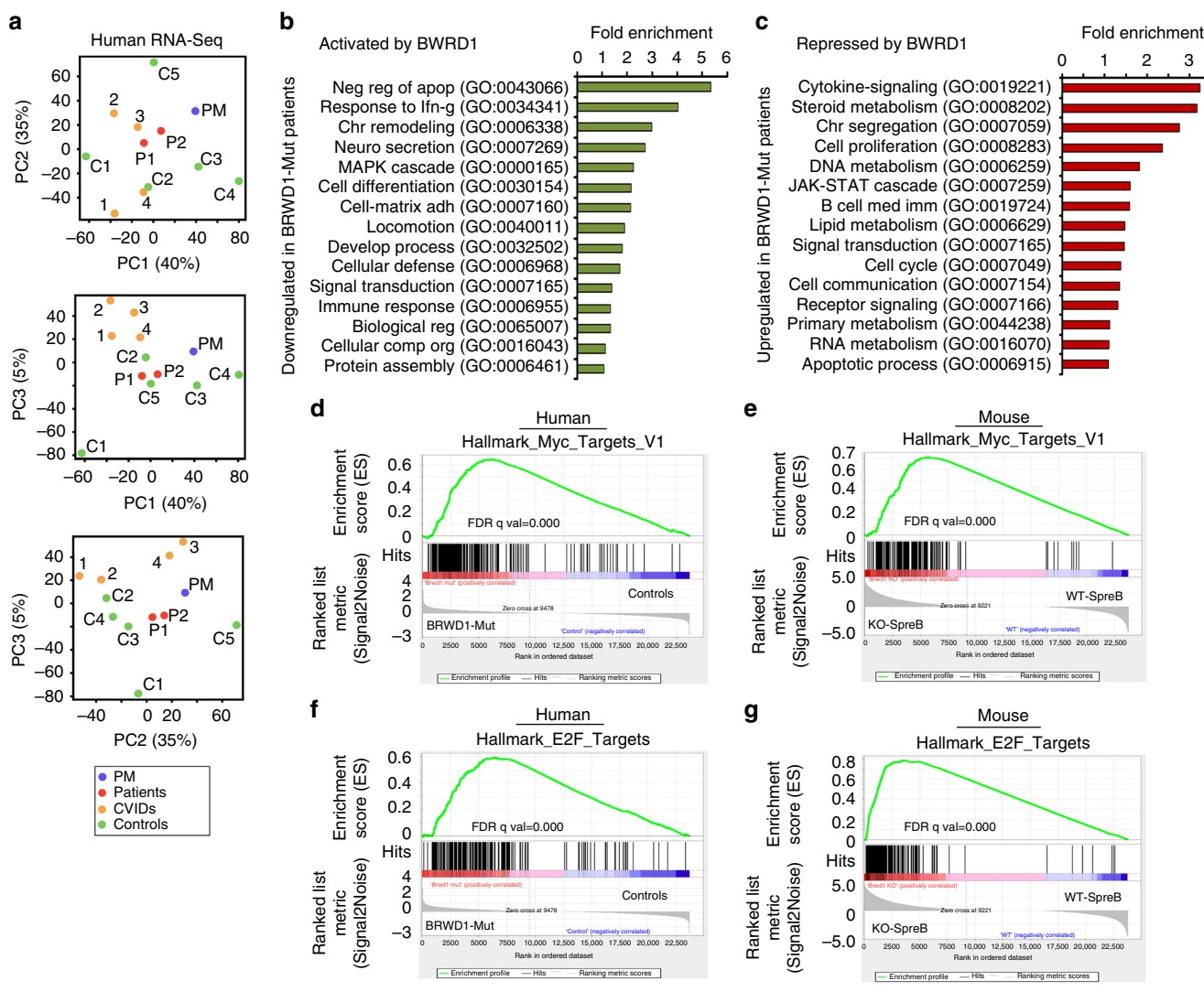

**Fig. 6** Human BRWD1 represses proliferative gene expression program. **a** Principal component analysis of mRNA expression in all the EBV-transformed human peripheral blood B cells from patients (P1 and P2), controls (C1-5), unrelated CVID patients (CVID1-4) and patients' mother (PM). Eigenvalue statistics for the PC analysis of RNA-Seq were reported in axes. **b, c** Ontology analysis of gene clusters with lower (**b**) or higher (**c**) transcript abundance in *BRWD1*-Mut hypogammaglobulinemic patient group (P1-2 than in control group, C1-5). Only terms with FDR < 5%, P < 0.05 and over log$_2$ 2 fold enrichment were reported. The numbers in the parenthesis indicate the associated pathway. **d, e** Gene set enrichment analysis (GSEA) of MYC pathway enriched in P1/2 relative to controls (**d**) and *Brwd1*$^{-/-}$ small pre-B cells compared to WT small pre-B cells (**e**). **f, g** Gene set enrichment analysis (GSEA) of E2F pathway enriched in *BRWD1*-Mut patient group relative to controls (**f**) and *Brwd1*$^{-/-}$ small pre-B cells compared to WT small pre-B cells (**g**)

We next performed RNA-Seq on EBV-transformed peripheral blood B cells from two patients (P1 and P2), their mother (PM) who carried the same *BRWD1* mutation but was healthy, five healthy controls (C1–C5), and four unrelated common variable immunodeficiency disease (CVID) patients (CVID1-4). Hierarchical clustering of gene expression (Fig. 5d) revealed that P1 and P2 had similar expression patterns and that these were very different from that observed in healthy controls. Expression in PM was intermediate between P1/2 and healthy controls. In contrast, CVID patients had heterogeneous distributions of expression likely reflecting different underlying casual mutations. Alignment of genes that were differentially expressed in P1/P2 and *Brwd1*$^{-/-}$ mice (Supplementary Figs. 4, 5) revealed similar patterns of expression that were different from those observed in either healthy controls or CVID patients.

**BRWD1 represses proliferation in human B cells.** Pairwise analysis between patient group (P1 and P2) and indicated

controls (Fig. 5e) revealed that at least 1100 genes were differentially expressed in any given comparison (*q* < 0.05). In contrast, 732 genes were differentially expressed between BRWD1 mutant patients and their mother. Principal component analysis (Fig. 6a) confirmed that mRNA expression in P1 and P2 was similar and closely related to PM. In contrast, expression in the healthy controls and CVID patients were widely distributed. As seen in mice, BRWD1 appeared to repress the expression of metabolic and cell cycle pathways while inducing expression of molecular pathways of activation and differentiation (Fig. 6b, c). Indeed, there was a similar enrichment in expression of MYC/Myc-induced genes (Fig. 6d, e) and E2F (Fig. 6f, g) target genes in P1–2 and *Brwd1*$^{-/-}$ cells.

**Human BRWD1 modulates accessibility at specific enhancers.** Examination of chromatin accessibility revealed that P1–2 cells had 25,691 accessible peaks, while healthy controls had less with 19,038 peaks (Fig. 7a and Supplementary Fig. 6a). Accessible

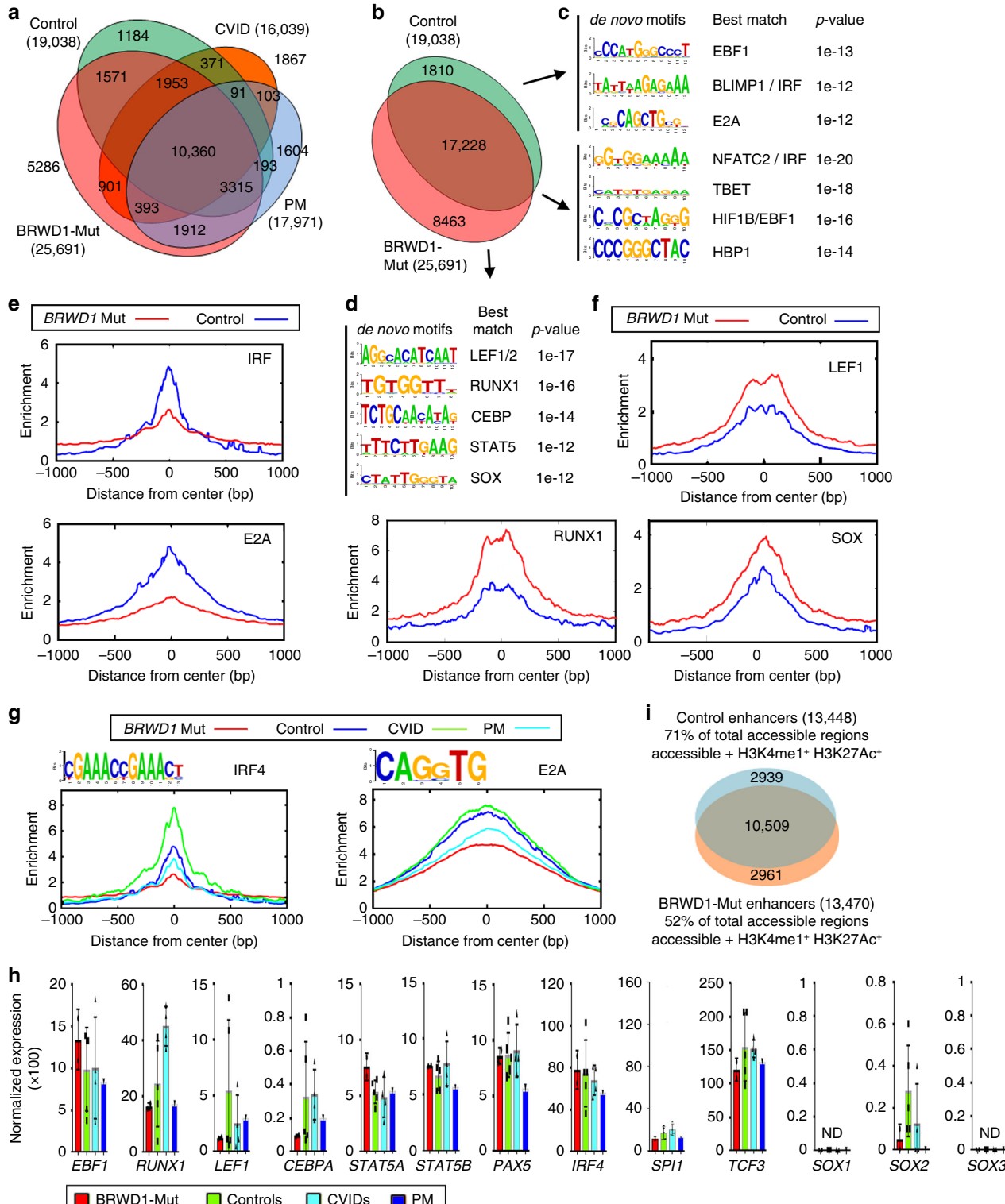

**Fig. 7** Human BRWD1 determines the genomic targets accessible to TFs. **a** Overlap of open chromatin peaks (ATAC-Seq) in EBV-transformed cells of patient group (P1 and P2), control group (C1-5), unrelated common variable immunodeficiency group (CVID1-4) and patients' mother (PM). **b** Overlap of open chromatin peaks (ATAC-Seq) in cells from indicated populations. For (**a**) and (**b**) total number of peaks for each population shown in parentheses with number in each Venn region indicated. **c**, **d** De novo DNA sequence motifs enriched in indicated areas of Venn diagram. **e**, **f** Average accessibility at transcription factors IRF4 and E2A (**e**) and LEF1, RUNX1 and SOX (**f**). Accessibility at indicated binding sites for patients (red) and control (blue) cells. **g** Average accessibility of IRF4 and E2A transcription factor binding sites in open chromatin regions of all the groups described in (**a**). **h** Quantitative normalized mRNA expression of transcription factors *EBF1, RUNX1, LEF1, CEBPA, STAT5A, STAT5B, PAX5, IRF4, SPI1* (PU.1),*TCF3 (E2A), SOX1, SOX2* and *SOX3* in EBV-transformed cells of *BRWD1*-mut hypogammaglobulinemia patients (P1 and P2), controls (C1-5), unrelated common variable immunodeficiency patients (CVID1-4) and PM along with Individual expression. Data are presented as average ± s.d. (n = 2). **i** Comparison of accessible enhancers (open chromatin+ H3K4me1+H3K27Ac+) between controls and *BRWD1*-Mut patients groups

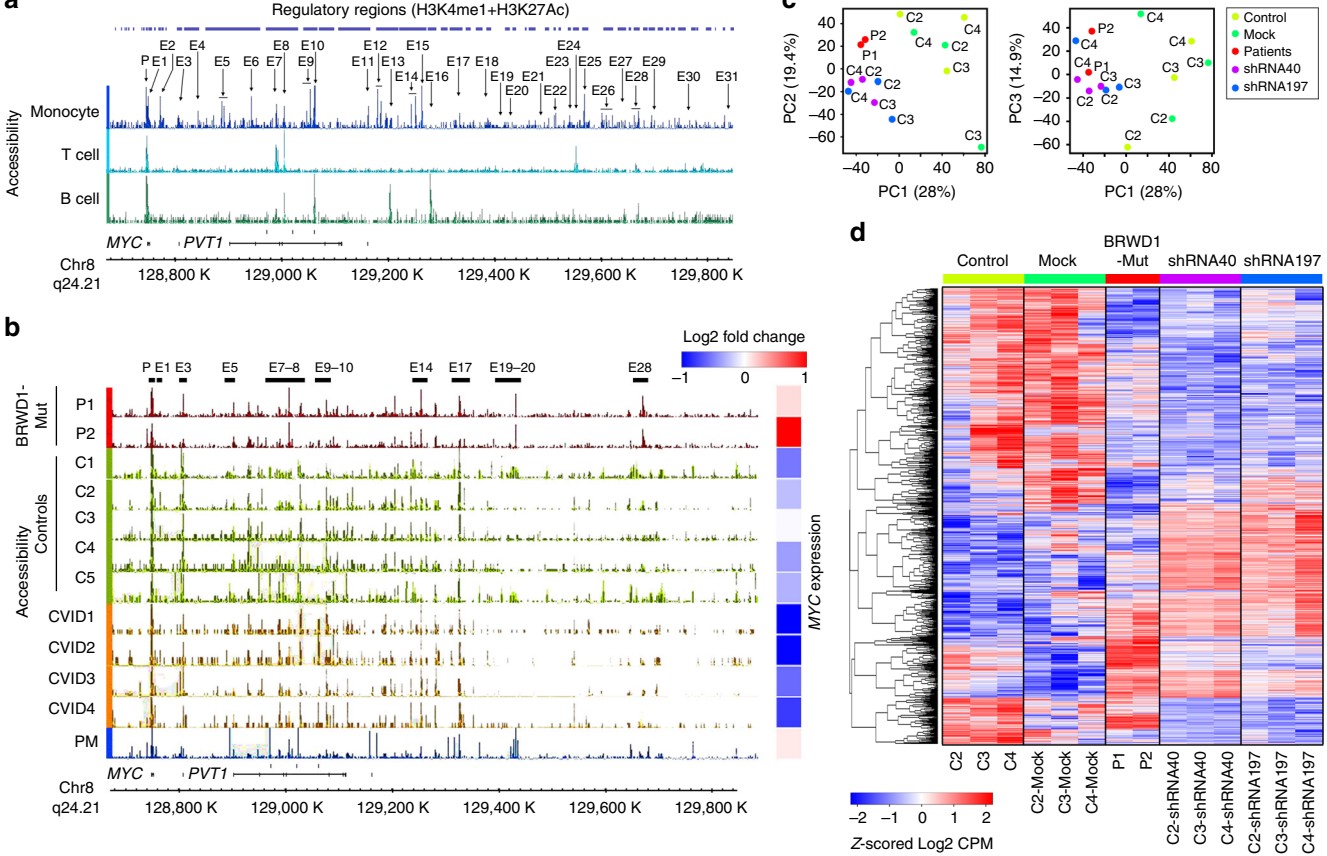

**Fig. 8** BRWD1 regulates *MYC* and other transcriptional programs in human peripheral B cells. **a** Identification of possible regulatory regions of human *MYC* locus using accessibility, H3K4me1 and H3K27Ac ChIP-seq data from monocytes, T cells and B cells (International Human Epigenome Consortium, IHEC). **b** Accessibility (open chromatin) profile of human *MYC* locus in BRWD1-Mut patients (P1 and P2), controls (C1-5), unrelated CVID patients (CVID1-4) and patients' mother (PM) with heatmap of *MYC* expression across each individual sample (right panel). **c** Principal component analysis of mRNA expression in EBV-transformed human peripheral blood B cells from healthy controls (C2, C3, and C5), patients (P1 and P2), mock-shRNA and BRWD1-specific shRNA40 and ShRNA197 expressing controls (C2, C3, and C4). Eigenvalue statistics for the PC analysis of RNA-Seq are reported in axes. **d** Hierarchical clustering of differentially expressed genes identified at *q* < 0.05 in the above **c** shRNA-expressing and non-expressing EBV-transformed control cell lines were shown

peaks in PM and CVID patients were slightly reduced compared with controls (17,971 and 16,039, respectively), with 10,360 peaks being shared between all groups (Fig. 7a and Supplementary Fig. 6a). In those accessibility peaks found in control but not P1-2 cells (Fig. 6b), there was an enrichment for accessibility of EBF1, IRF, and E2A TF binding sites (Fig. 7c and Supplementary Fig. 6b). In contrast, at those accessibility sites specific to P1–2 cells (Fig. 7b), there was an enrichment of LEF1/2, RUNX, CEBP, STAT5, and SOX binding sites (Fig. 7d and Supplementary Fig. 6b). Genome-wide accessibility at IRF and E2A motifs was diminished in P1/P2 B cells (Fig. 7e), while accessibility at LEF1, RUNX1, and SOX sites was increased (Fig. 7f). Comparison with IRF4 and E2A sites genome-wide in all groups demonstrated that accessibility was lowest in P1–2 and highest in the CVID patients (Fig. 7g). As was observed in $Brwd1^{-/-}$ mice, *BRWD1* mutations were not associated with substantial differences in expression of TFs critical for either lymphopoiesis or myelopoiesis (Fig. 7h). Additionally, BRWD1 preferentially regulated the enhancers (Fig. 7i), as was observed in $Brwd1^{-/-}$ mice (Fig. 3b). Examination of genes derepressed in P1–2 cells, and $Brwd1^{-/-}$ small pre-B cells, revealed that increased expression was associated with increased accessibility at specific enhancers (Supplementary Fig. 7a–f). The converse relationship between enhancer accessibility and gene expression was observed in genes induced by BRWD1 (Supplementary Fig. 8a–f).

The human *MYC* locus has 31 putative enhancers (Fig. 8a). Comparison with accessibility across the locus revealed that *BRWD1* mutations were specifically associated, in both P1–2 and PM, with enhanced accessibility at E7–8, E19–20, and E28 and increased *MYC* expression (Fig. 8b). Overall, these data indicate that in humans, *BRWD1* mutations are associated with transcriptional and epigenetic signatures similar to those observed in $Brwd1^{-/-}$ mice.

**BRWD1 is required for normal human B-cell transcription.** The above data indicate that mutations in *BRWD1* are associated with human B-cell immunodeficiency. Furthermore, these mutations appear to have an autosomal dominant pattern of inheritance with incomplete penetrance (Fig. 5). Therefore, to independently confirm that BRWD1 is biologically active in human B cells, we used two different short hairpin RNAs (shRNA40-RFP and shRNA197-RFP, Supplementary Table 6) targeting *BRWD1*. Retrovirus expressing each shRNA, or empty vector, was used to infect EBV-transformed peripheral blood B cells from three normal individuals (C2, C3, and C4). Infected cells were then sorted by flow cytometry and subjected to RNA-Seq. Initial analysis of the RNA-Seq data revealed that shRNA40 and shRNA197 diminished BRWD1 expression by 58% and 64%, respectively. Principal component analysis confirmed that mRNA

expression in patients (P1 and P2, red) was similar and closely related to *BRWD1*-specific shRNA-expressing controls (blue and purple). In contrast, expression in all the above populations was very different than either mock infected or healthy controls (Fig. 8c, green). Hierarchical clustering of gene expression (2786 genes by multi-group differential analysis between control, BRWD1-Mut, mock, shRNA40, and shRNA197 groups, and filtered the genes on *q*-value < 0.05) revealed that mock-transfected cells have similar expression patterns to C2, C3, and C4. However, control cells expressing *BRWD1*-specific shRNAs have similar expression patterns to P1 and P2, and these were very different from that observed in healthy controls or mock-transfected controls (Fig. 8d). The correlation between *BRWD1* shRNA and BRWD1-Mut cells was most striking for genes that were downregulated compared with controls. In toto, these data demonstrate that BRWD1 plays a critical role in normal human humoral immunity.

## Discussion

Our studies demonstrate that TF networks are not sufficient to regulate developmental programs of late B lymphopoiesis. In addition, expression and recruitment of BRWD1 to chromatin is critical for determining the accessibility of enhancers to TFs, such as IRF4. Together, regulated TF networks and BRWD1 both silence the transcriptional programs of early development and induce those required for *Igk* recombination and subsequent B-cell development. In humans, BRWD1 appeared to have similar critical functions highlighting the central and conserved function of BRWD1 in B lymphocytes.

Like the important TFs of B lymphopoiesis[1], the expression and recruitment of BRWD1 is dependent on stage-specific signals that restrict the activities of BRWD1 to a precise developmental context. The expression of *Brwd1* is repressed by STAT5 and therefore escape from IL-7R signaling is necessary for BRWD1 induction[31]. Subsequently, upon expression of pre-BCR, signals are activated that lead to the deposition of the histone acetyl and phosphorylation marks (H3K9acS10pK14ac) that recruit BRWD1 to chromatin[31]. At *Jk*, the necessary histone acetylation marks require E2A, which is downstream of the pre-BCR, and binds the *Igk* intronic enhancer (iEk)[11,31]. However, in many cases, BRWD1 is recruited to sites that are not enhancers, suggesting that recruitment is not solely dependent on pioneer TF binding. Furthermore, independent of histone acetylation, H3S10 can be directly phosphorylated by ERK and p38[31,40]. These observations suggest that the epigenetic landscape that recruits BRWD1 results from intergration of multiple developmentally restricted signaling pathways.

One of the remarkable functions of BRWD1 was the coordinated repression of *MYC* and MYC target genes. At the *Myc*/*MYC* locus, in both mice and humans, BRWD1 specifically repressed distal enhancers, which are associated with lineage-specific expression. This is in contrast to both BRD4[41] and Brg1[34], which induce *MYC* expression by opening specific enhancers. Indeed, MYC is thought to be a main oncogenic target of BRD4 and provides much of the rationale for targeting BRD4 therapeutically in various cancers[41,42]. Together, our studies and these published observations reveal another level of complexity in *MYC* regulation, in which functionally antagonistic epigenetic readers determine the expression.

BRWD1 has a predicted molecular weight of 263 kD and contains tandem BROMO domains and WD40 repeats[43]. However, it does not have strong homology with known catalytic domains. Therefore, it likely assembles a multimeric complex that then either opens or closes the chromatin. One component is likely SMARCA4 (Brg1), which co-immunoprecipitates with BRWD1[43] and is a component of the SWI/SNF chromatin remodeling complex[34,43]. The BRWD1 complex likely contains histone acetyltransferases as at *Jk*s, BRWD1 binding is associated with H4K16ac which plays a direct role in opening the chromatin[31,44]. Intriguingly, in regions opened by BRWD1, nucleosomes are positioned relative to repetitive GAGA DNA motifs[31], suggesting that the BRWD1 complex functions similarily to the GAGA factor Trithorax expressed in *Drosphilia*[45]. Interestingly, the sites repressed by BRWD1 are also enriched for GAGA motifs (M.R.C. and M.M., unpublished observation), suggesting some commonality in the mechanisms by which the BRWD1 complex both opens and closes the enhancers.

One the remarkable features of BRWD1 is that it regulated the enhancer accessibility over genomic distances of up to several mb. It is possible that this simply reflected an inability to detect small, biologically important, BRWD1–chromatin binding events. However, relaxing peak calling stingency did not change the fundamental relationship between BRWD1-bound sites and potential target genes. Therefore, our observations are not likely a consequence of experimental artifact. The mechanism by which BRWD1 regulates enhancer accessibility at a distance is unclear, although it likely requires three-dimenisonal interactions similar to those that enable the regulation of promoters by distal enhancers[16]. Further work is required to determine the interdependency of BRWD1 and chromatin three-dimensional structure in small pre-B cells.

In addition to determining the functions and importance of BRWD1, our studies reveal the extent of the transcriptional and epigenetic rewiring that occurs in small pre-B cells. This molecular reordering includes transition from proliferation, which selects for cells expressing competent immunoglobulin heavy chains, to a state conducive to *Igk* recombination. However, thousands of genes are differentially expressed indicating that the small pre-B cell transition involves more than just repressing cell cycle genes and inducing the core recombination machinery.

One aspect of early B lymphopoiesis, and the pre-B cell transition in particular, is the silencing of early developmental and even multipotential enhancers. Transit from the large pre-B-cell to small pre-B-cell stage is associated with a loss of almost 20,000 accessibility sites, with most of these being enhancers. In the absence of BRWD1, many of these enhancers become open. Furthermore, some enhancers associated with the multipotential pre-pro B-cell stage opened including those targeted by the myeloid TF CEBP. This suggests a role for BRWD1 in lineage specification. It is not clear if derepression of pre-pro B-cell enhancers reflects silencing in pre-B cells or a role for BRWD1 earlier in development where it is expressed at low levels. However, the first defect in B-cell progenitor numbers in *Brwd1*−/− mice is at the small pre-B-cell stage. Therefore, if BRWD1 plays a role in early lineage specification, it is not critical for early B-cell development.

Interestingly, many of the genes silenced by BRWD1 in small pre-B cells are re-expressed in germinal centers. Likewise, many BRWD1-induced genes are repressed. This suggests a general repurposing of developmental genes for the rapid proliferation and selection that occurs in germinal centers. It is possible that BRWD1 might play a role in modulating these GC transcriptional programs, as it is highly expressed in late GC B cells (M.R.C. and M.M., unpublished observation).

In humans, we identified three different *BRWD1* mutations associated with severe B lymphopenia and hypogammaglobulinemia. All three mutations are predicted to lead to expression of a truncated or mutant protein that could act as a dominant negative. This could explain why the B-cell deficiency observed in some patients was more severe than that observed in *Brwd1*−/− mice. *BRWD1* mutations occurred in 8% of screened patients

indicating that if is both a novel and relatively common cause of human immunodeficiency.

Recent large-scale whole-genome sequencing studies, including the 1000 genomes project (http://www.internationalgenome.org), demonstrate that healthy individuals can have large numbers of genome-associated genetic variants without manifesting disease. Most monogenic, and essentially all polygenic, human diseases display reduced penetrance[46]. Several factors contribute to reduced penetrance including gender, environment, and the presence of modifier genes. In the family with the *BRWD1* mutation, the two affected individuals were both male and had grown up with similar environmental exposures. All three *BRWD1* mutant individuals we studied only shared 50% genetic relatedness, which means one or more might have different modifier genes. Furthermore, although the mother did not manifest disease, she did share some molecular features associated with the *BRWD1* mutation in the brothers and manifest in *Brwd1^{-/-}* mice. Therefore, the reduced disease penetrance in mother of the patients is not unexpected. Furthermore, our shRNA studies directly demonstrate that BRWD1 is required for normal transcription in human peripheral B cells. Our data provides the first example of a human immunodeficiency associated with mutations of an epigenetic reader gene[47].

There was remarkable concordance in the genes dysregulated in murine *Brwd1^{-/-}* small pre-B cells and *BRWD1* mutant human transformed peripheral B cell lines. We postulate, this reflects the strong and central role that BRWD1 plays in B-cell function. It also suggests that BRWD1 might plays similar roles in both the bone marrow and periphery. Indeed, BRWD1 is highly expressed in follicular B cells and splenic follicular architecture is disrupted in *Brwd1^{-/-}* mice (M.C. and M.M., unpublished observations). Futher studies are needed to determine how BRWD1 contributes to peripheral humoral immunity.

Our studies provide an integrated model of late B lymphopoiesis. As B-cell progenitors transit to the small pre-B-cell stage, early developmental TFs are repressed and those critical for late development induced. However, for these TFs to direct late lymphopoiesis, critical enhancers must be available for binding. This is the function of BRWD1. Both TF network expression and BRWD1 function are dependent upon a complex integration of intrinsic and extrinsic developmental signals that ensure that only those cells most fit to express a functional BCR proceed along the developmental pathway.

## Methods

**Mice**. WT and *Brwd1^{-/-}* mice were housed in the animal facilities of the University of Chicago. Mice were used at 6–12 weeks of age, and experiments were carried out in accordance with the guidelines of the Institutional Animal Care and Use Committee of the University of Chicago.

**Analysis of bone marrow B-cell progenitors**. Bone marrow (BM) was collected from WT or *Brwd1^{-/-}* mice, and cells were resuspended in the staining buffer (3% (vol/vol) FBS in PBS). Erythrocytes were lysed, and cells were stained with anti-CD11c (HL3), anti-NK1.1 (PK136), anti-TCRβ (H57-597), anti-CD71 (C2), anti-Ter119 (TER-119), anti-Mac-1 (M1/70), anti-Gr-1 (RB6-8C5), anti-CD34 (RAM34) (1:200), anti-Sca1 (Ly-6A/E, D7) (1:200), anti-cKit (CD117, 2B8) (1:200), anti-Flt3 (CD135, A2F10.1) (1:200), anti-IL7Rα (CD127, SB/199) (1:100), anti-CD4 (H129.19), anti-CD8 (53-6.7), anti-CD25 (IL2Rα, 7D4), anti-CD44 (IM7), anti-TCRγδ (GL3), anti-CCR9 (CW1.2), anti-CXCR4 (2B11) (1:100), anti-CD43 (S7), IgM (R6-60.2), IgD (11-36), anti-CD19 (1D3), anti-B220 (RA3-6B2l), and anti-CD93 (AA4.1), CD21 (7G6), CD23 (B3B4) (all from BD Biosciences or Biolegend). Antibodies were directly coupled to fluorescein isothiocyanate, phycoerythrin, phycoerythrin–indotricarbocyamine, allophycocyanin, eFluor 450, or biotin and were used at 1:400 dilution, except otherwise mentioned. Pre-pro B cells (Lin^{neg}CD19^{neg}B220^{+}IgM^{neg}), pro-B cells (Lin^{neg}CD19^{+}B220^{+}CD43^{+}IgM^{neg}), large pre-B cells ((Lin^{neg}B220^{+}CD43^{neg}IgM^{neg}FSC^{hi}), small pre-B cells (Lin^{neg}B220^{+}CD43^{neg}IgM^{neg}FSC^{low}), and immature B cells (Lin^{neg}B220^{+}CD43^{neg}IgM^{+}) were isolated by cell sorting with a FACSAriaII (BD).

**Quantitative PCR analysis**. Total cellular RNA was isolated with an RNeasy kit (Qiagen), and RNA was reverse-transcribed with SuperScript III reverse transcriptase (Invitrogen). For quantitative PCR, a total volume of 25 μl containing 1 μl cDNA template, 0.5 μM of each primer (Supplementary Table 6), and SYBR Green PCR Master Mix (Applied Biosystems) was analyzed in quadruplicate. Gene expression was analyzed with an ABI PRISM 7300 Sequence Detector and ABI Prism Sequence Detection Software version 1.9.1 (Applied Biosystems). Results were normalized by division of the value for the unknown gene by that obtained for *B2m*.

**Short hairpin RNA**. Oligonucleotides of shRNA specific for human BRWD1 (97 bases; targeting sequences are presented in Supplementary Table 6) were designed according to the RNAi Consortium criteria and software (Broad Institute; http://www.broadinstitute.org/rnai/public/) and through the use of Ravi Lab Resources (http://katahdin.cshl.org/siRNA/RNAi.cgi?type=shRNA). These shRNA oligonucleotides were cloned into a RFP-expressing retroviral vector based on microRNA miR30 (Ref[31]).

**Flow cytometry**. Analysis of total intracellular IRF4 was performed in flow sorted small pre-B cells (Lin^{neg}B220^{+}CD43^{neg}IgM^{neg}FSC^{low}) from WT and *Brwd1^{-/-}* mice bone marrow as described previously[10]. Briefly, WT and *Brwd1^{-/-}* small preB cells were fixed and permeablized using True-Nuclear™ Transcription Factor Buffer Set (Biolegend, 424401). Cells were then stained intracellularly with anti-IRF4-AF647 (Biolegend, clone IRF4.3E4, 646408) or isotype control (Biolegend, 400418), and analyzed by flow cytometry.

**ChIP-sequencing**. Chromatin from flow sorted small pre-B cells (20 × 10^6 or 2 × 10^6 for low DNA samples from WT and *Brwd1^{-/-}* mice was used for each ChIP experiment with anti-IRF4 (M-17, Santa Cruz Biotechnogy, sc-6059 ×, lot#J2015) and anti-BRWD1 (E-15, Santa Cruz Biotechnogy, sc-83517, lot J1508) antibodies described above. DNA libraries were prepared from the sheared chromatin (200–600 bp). Libraries were sequenced on the Illumina Hiseq2500. The sequences were aligned to the mm9 reference genome (National Center for Biotechnology Information build mm9_NCBI_build_37.1) with Bowtie alignment software[48] and only reads with unique matches were retained.

**ChIP-Seq peak calling and motif analysis**. Peaks for ChIP-Seq samples were called using MACS2 at a *p*-value threshold of 10^{-5} or 10^{-7} as done previously[31]. Normalized bedgraph tracks were generated using the SPMR flag, and converted to bigWig using UCSC tool bedGraphToBigWig. Peaks with a score > 5 were retained. Peak groups were generated by considering overlapping peak regions by at least 10 bp. HOMER software (hypergeometric optimization of motif enrichment) for de novo motif discovery and next-generation sequencing analysis was used for new prediction of motifs in the peaks. Additionally, de novo motif searches were performed independently on each peak group using MEME asking for the top 10 motifs.

For further motif analysis and DNA footprinting, peaks were recalled at a *p*-value threshold of 10^{-7}. We then searched for the identified de novo motifs in each of these peak groups using MAST counting both the total number of hits for the motif, and the fraction of sequences with at least one hit for the motif.

**RNA-sequencing and analysis**. RNA was isolated with the Qiagen RNeasy kit (Qiagen). Concentration and purity was analyzed on a Nanodrop 1000 Spectrophotometer. mRNA was isolated by oligo-dT beads and library was prepared using standard Illumina library protocol (Kit: RS-122-2101 TruSeq® Stranded mRNA LT-SetA). Libraries were sequenced on the Illumina Hiseq2500.

All raw sequencing data was quality trimmed to a minimum phred score of 20 using trimmomatic[49]. QC-passed reads were filtered against a reference of mouse rRNA sequences using bowtie2 (v.2.2.4)[48] with default parameters, then aligned to the mm9 reference genome in a splice-aware manner using the STAR aligner. Gene expression was quantified using cuffquant, and normalized expression levels and differential expression levels were generated with cuffnorm and cuffdiff, respectively (version 2.2.1)[50]; default parameters were used, except that multi-read correction was turned on, library type was set to "fr-secondstrand", and fragment bias correction was performed. UCSC gene annotations for mm9 were used for transcriptome mapping and quantification. Differential expression statistics (fold-change and *p*-value) were computed using edgeR, on raw expression counts obtained from quantification. *P*-values were adjusted for multiple testing using the false discovery rate (FDR) correction of Benjamini and Hochberg.

**Assay of transposase-accessible chromatin using sequencing**. ATAC-Seq was performed as recently described[31]. Briefly, to prepare nuclei, respective mouse or human cells (1 × 10^5) were centrifuged at 500 *g* for 5 min, which was followed by a wash with ice-cold PBS and centrifugation at 500 *g* for 5 min. Cells were lysed using cold lysis buffer (10 mM Tris-HCl, pH 7.4, 10 mM NaCl, 3 mM MgCl_2 and 0.1% IGEPAL CA-630). Immediately after lysis, nuclei were spun at 500 *g* for 10 min at 4 °C and supernatant removed. The nuclei pellet was resuspended in the transposase reaction mix (25 μL 2X Tagment buffer, 2.5 μL Tagment DNA enzyme

(Illumina, FC-121-1030) and 22.5 μL nuclease-free water). The transposition reaction was carried out at 37 °C for 30 min. Directly following transposition, the sample was purified using a Qiagen MinElute kit. Following purification, we amplified library fragments using Nextera PCR Primers (Illumina Nextera Index kit) and NEBnext PCR master mix (New England lab, 0541) for a total of 10–12 cycles. The libraries were then purified using a Qiagen PCR cleanup kit.

The amplified, adapter-ligated libraries were size selected using Life Technologies' E-Gel® SizeSelect™ gel system in the 150–650 bp range. The size-selected libraries were quantified using the Agilent Bioanalyzer and via q-PCR in triplicate using the KAPA Library Quantification Kit on the Life Technologies Step One System. Libraries were sequenced on the Illumina Hiseq2500 system.

**QC and DNA alignment**. All raw sequence data was quality trimmed to a minimum phred score of 20 using trimmomatic[49]. Alignment to reference genome mm9 was done with BWA[51]; for ATAC-Seq data, read pairs where one pair passed quality trimming but the other did not were aligned separately and merged with the paired-end alignments. PCR duplicates were removed using Picard Mark Duplicates and alignments with an edit distance greater than two to the reference, or that were mapped multiple times to the reference, were removed.

**ATAC-Seq analysis**. ATAC-Seq analysis followed the procedure described previously[31]. Read alignments were first adjusted to account for TAC transposon binding: +4 bp for +strand alignments, -5bp for -strand alignments. The open chromatin enrichment track was generated by first creating a bedGraph from the raw reads using bedtools genomcov, then converted to bigWig using UCSC tool bedGraphToBigWig; tracks were normalized by the sum of alignment lengths over 1 billion. The start position track was generated by taking just the first base of the alignment for +strand alignments or the last base of the alignment for - strand alignments, then creating bedGraph and bigWig tracks as for the open chromatin; tracks were normalized to the alignment count over 1 million. Open chromatin peaks were called using Macs2 with --nomodel set and no background provided; peaks with a score >5 were retained. Furthermore, we used the peak calling results as a guide to identify regulatory elements, then quantified enrichment in these regulatory elements and ran differential analysis to compare samples. Reproducibility was factored into the differential statistics that we calculated using estimates of dispersion.

For nucleosome positioning, properly paired alignments were filtered by their fragment size. Fragment sizes <100 bp were considered nucleosome free and replaced with a single BED region, and used as a background. Sizes between 180 and 247 bp were considered mononucleosome and replaced with a single BED region; sizes between 315 and 473 bp were considered dinucleosomes and replaced with two BED regions, each spanning half the overall fragment length; and sizes between 558 and 615 bp were considered trinucleosomes and replaced with three BED regions, each spanning one third of the overall fragment length; the mononucleosome, dinucleosome, and trinucleosome regions were concatenated and used as the nucleosome signal. The resulting BED regions were analyzed using DANPOS[52] with the parameters –p 1 –a 1 –d 20 ––clonalcut 0 to identify regions enriched or depleted for nucleosomes.

DNA footprinting data were obtained by combining bigWig enrichment tracks for ChIP-Seq and ATAC-Seq data over specified BED regions (combinations of peak calls or motif hits). ChIP-Seq enrichment data were generated by MACS2, as described above. Open chromatin enrichment data from ATAC-Seq were generated from the read-adjusted alignments using custom scripts, normalized to reads per million alignments, and nucleosome positioning enrichment data were obtained from DANPOS[52]. DNA footprinting scores were averaged over 10 bp bins from enrichment tracks, using custom scripts.

For correlations between signals, the UCSC genome browser's bigWigToWig tool was used to extract profiles for the *Igk* loci from each of the enrichment tracks. Different enrichment profiles were compared using both the Pearson and Spearman correlation coefficients; the latter was included to prevent regions of very high enrichment from dominating the correlation.

**Peak gene association analysis**. Differential expression was generally described by two numbers, the log fold-change and the FDR-corrected *p*-value. To capture both the size of the change and significance, these two criteria were merged together into a single statistic: log fold-change × -log FDR. For ATAC-Seq or BRWD1-ChIP-Seq to gene expression (RNA-seq) associations the non-overlapping distance cutoffs (bp between the peak position and the TSS) were used to see how the effect depends on distance to TSS. The ranges analyzed were 0–0 kb, 0–5 kb, 5–20 kb, 20–100 kb, 100–200 kb, 200–500 kb, and 500 kb–5 mb. The figures were presented to describe how a subset of genes on the basis of a certain set of peaks affects the distribution of gene expression levels. Differential histogram plotted the probability distribution of differential expression for different peak set. The Q–Q plotted directly plots the quantiles of the background distribution against the quantiles for each peak set. For both plots the overall distribution from all genes were set as background.

In Q–Q plots demonstrating relationship between accessibility (ATAC-Seq) and mRNA expression (RNA-Seq) in WT and *Brwd1*$^{-/-}$ small pre-B cells, differential

expressions were described by merging the log fold-change and the FDR-corrected *p* value in a single statistic (log fold-change x –log FDR) to capture both the magnitude of change and significance. Quantiles of overall background distribution (Bg, black) and quantiles for expression associated with increased (up, red), decreased (down, blue), or unchanged (noc, green) mRNA expression (*Brwd1*$^{-/-}$ vs. WT) against chromatin accessibility (*Brwd1*$^{-/-}$ vs. WT) as a function of distance from TSSs (0–5, 5–100, 100–200, and 200–500 kb).

**PBMC Isolation and generation of EBV-B cell lines**. PBMCs from patients and controls were isolated using Ficoll–Paque gradients. Epstein–Barr virus (EBV) was generated from the blood of the cotton-top tamarin monkey peripheral blood lymphocytes (Sigma, B95-8). EBV virus (1 ml) was added to 3 ml of $10^6$ PBMCs in RPMI 1640 with 20% FBS. Cells were cultured for two weeks and then stored at −80 °C.

**Sanger sequencing**. Genomic DNA was extracted from PBMCs using the ArchivePure DNA extraction kit (5Prime). PCR amplification on genomic DNA was performed using HotStarTaq (Qiagen) according to the manufacturer's instructions. Sanger sequencing was performed by Macrogen (https://www.macrogenusa.com/) and analyzed using 4Peaks (http://nucleobytes.com/4peaks/). Primers were listed In Supplementary Table 7.

**Whole-exome sequencing**. Genomic DNA from peripheral blood mononuclear cells was extracted and sheared with a Covaris S2 Ultrasonicator. An adapter-ligated library (Illumina) was generated, and exome capture was performed with SureSelect Human All Exon 71 Mb kit (Agilent Technologies). Massively parallel whole-exome sequencing was performed on a HiSeq 2000 or 2500 (Illumina), which generates 72-base or 100-base reads. We used the Genome Analysis ToolKit (GATK) best-practice pipeline to analyze our WES data[53]. Reads were aligned to the human reference genome (hg19) using Maximum Exact Matches algorithm in Burrows–Wheeler Aligner (BWA)[51]. PCR duplicates were removed using Picard tools (http://picard.sourceforge.net/). The GATK base quality score recalibrator was applied to correct sequencing artifacts. GATK HaplotypeCaller was used to identify substitution and insertion/deletion variant calls, respectively. All variants with a Phred-scaled SNP quality ≤30 were filtered out. We also filtered out variants with a coverage depth (CD) <8, a genotype quality (GQ) <20, or a minor-read ratio (MRR) <20% using a homemade script. Variant annotation was performed with SnpEff (http://snpeff.sourceforge.net/).

**Statistical analysis**. Data were analyzed with the unpaired *t*-test and analysis of variance, followed by the test of least-significant difference for comparisons within and between groups. All categories in each analyzed experimental panel were compared *P*-values below 0.05 were considered significant. All *P*-values below 0.001 were rounded to facilitate comparisons of results.

## Data availability

The data generated during and/or analyzed during the current study are available from the corresponding author(s) on reasonable request. GEO accession codes for publicly available data sets: Small pre-B BRWD1 ChIP-Seq (GSE63302); Pro-B c-Myc ChIP-seq GSE40173; Pro-B H3K4me1 ChIP-seq (GSE21978), pro-B (CD19+) H3K27ac (GSE24164), CH12 H3K4me1 ChIP-seq (ENCSR000DHQ), CH12 H3K27ac ChIP-Seq (GSE31039), Bone Marrow H3K4me1 (GSE31039), Bone Marrow H3K4me3 (GSE31039), Bone Marrow H3K27ac (GSE31039). Human B cell, T cell and Monocyte ATAC-Seq, H3K4me1, and H3K27Ac ChIP-Seq data used were shared by the McGill Epigenomics Mapping Centre and it is available from the European Genome-phenome Archive of the European Bioinformatics Institute (http://epigenomesportal.ca/edcc/data_access.html) and International Human Epigenome Consortium (IHEC) (http://epigenomesportal.ca/ihec/)[54]. The sequences, ATAC-Seq for WT pre-proB, WT pro-B, WT large pre-B, WT small pre-B, *Brwd*1$^{-/-}$ small pre-B and Immature B, RNA-Seq for WT pro-B, WT small pre-B and *Brwd1*$^{-/-}$ small pre-B, IRF4 ChIP-seq for WT and *Brwd1*$^{-/-}$ small pre-B, RNA-Seq and ATAC-Seq for human EBV-transformed peripheral blood B cells from patients (P1 and P2), patient mother (PM), healthy controls (C1-5), four unrelated Combined Immunodeficiency patients (CVID1-4) and RNA-Seq for mock-shRNA and human BRWD1-specific shRNA40 and shRNA197 expressing healthy controls C2, C3 and C4 reported in this paper has been deposited in the GenBank database (accession no GSE103057).

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

## Acknowledgements

We thank M. Olson and D. Leclerc for cell-sorting services; and the ImmGen Consortium for data assembly. This work is supported by the US National Institutes of Health Grants GM088847, GM101090 and U19 AI082724 (to M.R.C.), AI120715-02, AI128785-01A1 (to M.R.C. and M.M.), T32-GM007280 (to P.M.), AI101093, AI086037, AI48693 (to C.C.-R.), T32GM007281 (K.C.M.), UL1TR002003 (M.M.-C.), and The Jeffry Model Foundation grants (to P.M. and C.C.-R.).

## Author contributions

M.M. and M.R.C. designed the experiments; M.M. carried out and analyzed most of the experiments; M.M. and M.M.-C. analyzed the high throughput sequencing data; P.M. and C.C.-R. generated the peripheral B cell lines from hypogammaglobulinemic patients and healthy controls; M.V. assisted in flow cytometry; D.E.K., K.C.M., S.K., and M.K.O.

assisted M.M. in analyzing some data. M.M. and M.R.C. oversaw the entire project and wrote the final paper.

## Additional information

**Competing interests:** The authors declare no competing interests.

