## [Peer Review file · Nature Communications]

Reviewers' comments:

Reviewer #2 (B development, TF regulation)(Remarks to the Author):

In the revised manuscript, the authors included new experiments that address some of the previous concerns. In particular, the authors show that IRF4 protein levels are similar in small pre-B cells isolated from WT and *Brwd1*^{-/-} bone marrow. In addition, the authors performed a principal component analysis of mRNA expression in EBV-transformed cells from healthy controls, patients and BRWD1-knockdown cells, and they show differential gene expression in BRWD1-specific shRNA expressing control cell lines. Finally, the authors discuss the reduced disease penetrance of the BRWD1 mutation.

Together, these data provide additional evidence for the authors' conclusions and for the role of BRWD1 in human B cells. I agree with the authors that some of the suggested experiments, including Hi-C-Seq, are technically challenging and go beyond the scope of the current study. The revised manuscript has improved and will be of general interest.

Reviewer #3 (BCR repertoire generation, epigenetic; mediating for the original Nature Immunology Reviewer #1 who did not respond to my invitation)(Remarks to the Author):

Reviewer #1 has raised several valuable concerns, as follows:

"The data can't explain the mechanism by which BRWD1 switches between specific targets, controls their expression or alters chromatin accessibility"

>> The authors did not address this important concern. Rather, they argued that "defining how BRWD1 regulates enhancers genome-wide will take a great deal of work and is beyond the scope of this current manuscript". In a previous publication (*Nat. Immunol.* 16:1094–1103, 2015), the same authors showed that genome-wide BRWD1 binding was associated with enhanced DNA accessibility and enhanced RAG recruitment. In the *Nat. Commun.* manuscript currently under review, they show that (Fig. 1a and Supplementary Table 2) over 7,000 genes were dysregulated in *Brwd1*^{-/-} small pre-B cells with 3901 genes showing increased expression and increased chromatin accessibility, while 3405 genes showed decreased expression. This suggests that BRWD1 regulates chromatin accessibility at specific targets and controls their expression in different ways (up and/or down). Understanding how BRWD1 switches between specific targets and functions is critical to fully define the role of BRWD1 in the regulation of the epigenetic landscape of late B lymphopoiesis.<<

"The discrepancies between the number of the direct BRWD1 binding sites and the affected gene loci is not addressed".

>> This point was ignored by the authors.<<

"The genetic complementation experiment using the wild-type of mutant forms of BRWD1 will be exceptionally useful to address the mechanism of the BRWD1 function."

>>Due to technical difficulty of the genetic complementation experiment, the authors chose to address this point in a different way – knocking down BRWD1 with shRNAs in EBV-transformed cells from a normal individual and comparing gene expression in these cells to BRWD1^{-/-} cells. These experiments did provide some useful information regarding the role of BRWD1 in normal human B cell biology and function.<<

"On a more technical note, I am not sure about the number of the independent ATAC-seq experiments and related statistical analysis."

>>The author satisfactory addressed this point.<<

"Given the scope of BRWD1 impact on gene expression, I am also concerned about the gross impact of BRWD1 on B lineage division and growth. It's easy to imagine that any significant changes in the dynamic of the mutant cell division, differentiation will preclude the comparison with the wild-type cells due to the non-matching state of differentiation/division etc."

>>Given that expression of over 7,000 genes was altered in BRWD1^{-/-} small pre-B cells (Figure 1a), and BRWD1 is required for normal small pre-B cell development (as shown in the authors' 2015 Nat. Immunol paper), it is legitimate to believe that cell division and differentiation could be altered in BRWD1 mutant cells. Alterations in cell division and differentiation could directly or indirectly impact chromatin accessibility in certain loci. Although BRWD1 may directly positive or negatively regulates enhancer accessibility, one cannot rule out the possibility that differences in chromatin accessibility in BRWD1 knockout or mutant cells, as compared to wild-type cells, may partially due to the non-matching state of differentiation/division, etc. This point has not been addressed experimentally nor has it been discussed in the revised manuscript.<<

In conclusion, the authors' failure to address the following issues:

"...the mechanism by which BRWD1 switches between specific targets, controls their expression or alters chromatin accessibility..."

"The discrepancies between the number of the direct BRWD1 binding sites and the affected gene loci is not addressed"

"the gross impact of BRWD1 on B lineage division and growth... "

Greatly reduces the significance of this work and questions the suitability of the manuscript for publication in Nature Communications.

Reviewer #2: In the revised manuscript, the authors included new experiments that address some of the previous concerns. In particular, the authors show that IRF4 protein levels are similar in small pre-B cells isolated from WT and Brwd1^{-/-} bone marrow. In addition, the authors performed a principal component analysis of mRNA expression in EBV-transformed cells from healthy controls, patients and BRWD1-knockdown cells, and they show differential gene expression in BRWD1-specific shRNA expressing control cell lines. Finally, the authors discuss the reduced disease penetrance of the BRDW1 mutation.

Together, these data provide additional evidence for the authors' conclusions and for the role of BRWD1 in human B cells. I agree with the authors that some of the suggested experiments, including Hi-C-Seq, are technically challenging and go beyond the scope of the current study. The revised manuscript has improved and will be of general interest.

Response: We are pleased that the reviewer feels that our paper is improved and suitable for publication.

Reviewer #3 raised three concerns that are addressed below:

Comment#1: "The data can't explain the mechanism, by which BRWD1 switches between specific targets, controls their expression or alters chromatin accessibility"

Response: The reviewer thought that without an understanding of how BRWD1 regulated enhancer accessibility, the impact of the paper was significantly diminished. They then referenced our 2015 Nature Immunology paper as an example a sufficiently mechanistic body of work. We are pleased that the reviewer appreciates our previous efforts. However, the two papers are not comparable. The primary focus of the 2015 paper was *Igκ* recombination and how this was facilitated by BRWD1. By focusing on one gene, specifically one small region of the gene (*Jκ*), at one developmental stage, we could propose a specific mechanism. In contrast, the current paper focuses on the genome-wide importance and functions of BRWD1. In this manuscript, we clearly

provide a precise function for BRWD1 that we explore in great detail across several stages of B lymphopoiesis. Furthermore, as demonstrated in our extensive studies of *Myc* and *Myc* targets (Figure 3e and entire Supplementary Figure 3), we provide a broad and coherent mechanistic understanding of how BRWD1 is coordinating the whole *Myc*-regulated program of transcription. Furthermore, we demonstrate that BRWD1 dictates which enhancers IRF4 can bind genome-wide (Figure 4). Across 7 figures and 52 panels, we provide a very mechanistic picture in which BRWD1 orchestrates the opening and closing of developmentally regulated enhancers to critical transcription factors.

Our paper does more than just define the genome-wide functions of BRWD1. We demonstrate the importance of BRWD1 in human humoral immunity. Using three independent lines of investigation, whole exome sequencing, extensive characterization of B cell lines from patients harboring BRWD1 mutations and shRNA-targeting of BRWD1 in normal patient B cell lines, we demonstrate that BRWD1 is critical for normal human B cell function. Those patients with BRWD1 mutations have severe B cell lymphopenia, hypogammaglobulinemia and recurrent infections. Remarkably, the transcriptional and epigenetic state of B cell lines from patients was similar to that of B cells isolated from *Brwd1*^{-/-} mice. Therefore, in both mice and humans, BRWD1 plays a critical role in shaping B cell enhancer landscapes. We provide a cohesive and comprehensive picture of how BRWD1 contributes to B lymphopoiesis and the importance of these BRWD1-mediated mechanisms in both mice and humans. Our work is a substantial contribution to the global knowledge of mechanisms of both murine and human B cell lymphopoiesis.

Unfortunately, Reviewer #3 does not suggest specific experiments, making it difficult for us to address their concerns. It is possible that Reviewer #3 would be satisfied by experiments involving Hi-C Seq. However, obtaining enough cells for high-resolution Hi-C from a rare population in a B lymphopenic host would be very challenging. Repeated sorting over months and many mice (100+) would be required. Reviewer #2 agreed that it was not reasonable to expect such experiments in this current manuscript. Furthermore, such studies would not provide a mechanism. They would only provide an observation from which mechanisms could be hypothesized. Additional experiments would be required that are beyond the reasonable scope of this paper.

The mechanisms by which BRWD1 regulates enhancer accessibility are likely to be complicated and will require years to elucidate. The history of Aire is illustrative of what is being asked of us. Original papers described its function in mice (Anderson et al, Immunity, 2006; Giraud et. al., Nature, 2006) and importance in humans (Anderson MS et. al., Science, 2003). Only, recently has the molecular mechanisms by which AIRE mediates transcription become apparent (Meredith M, et. al., Nat Immunol, 2015; Bansal K et. al., Nat Immunol, 2017; Koh AS et. al., Nat Immunol, 2018). The reviewer is asking for this whole arc of investigation in a single paper, which we feel is an unreasonable request. Furthermore, such additional data may serve only to distract from the important points the current manuscript makes.

Comment #2: "The discrepancies between the number of the direct BRWD1 binding sites and the affected gene loci is not addressed".

Response: The concern raised by the Reviewer #3 was that there was an apparent discrepancy between the number of direct BRWD1 binding sites and the affected gene loci. However, there is no discrepancy. As demonstrated in Figures 2e-h, we explored at depth the ability of BRWD1 to regulate enhancer accessibility over distance of up to 5mb, which accounts for the apparent discrepancy.

However, the editor was felt we had not adequately discussed how BRWD1 could regulate accessibility at a distance including the possibility that this observation could arise from differential sensitivities of the methods used. To address this latter concern, we performed an additional analysis. It was possible that we over-estimated the distance between BRWD1 and regulated chromatin because we under-estimated the number of BRWD1-bound sites. Therefore, we relaxed the criteria by which we called peaks in our ChIP-Seq data set resulting in an increase in apparent BRWD1-bound sites from approximately 2000 to over 9000. Remarkably, this had no appreciable effect on the distance relationship between BRWD1 and potential target genes. This new analysis is provided in Figure 2f and page 8 of results (underlined in revised manuscript). This result is discussed on page 16 (underlined).

Comment #3: "the gross impact of BRWD1 on B lineage division and growth... "

Response: The concern raised by the reviewer is that our observations could reflect a general perturbation of B cell development leading us, by error, to sample an earlier stage of B lymphopoiesis in *Brwd1*^{-/-} mice. Certainly, with BRWD1 playing such a wide-ranging role in B lymphopoiesis this was a concern. However, we investigated the impact of BRWD1 on B lineage division and growth in our previous published paper (Mandal et. al. Nat Immunol 2015). The relevant data can be found in Supplementary Figures 3c and 3d. Furthermore, additional data presented in the current paper rigorously excludes the possibility that the observed phenotype reflects globally disordered development and sampling error. As demonstrated in Figures 3 and 4, very early B cell development enhancers open in *Brwd1*^{-/-} small pre-B cells. These enhancers are the targets of the transcription factors of early development. A particularly illuminating example is that of the *Myc* locus (Figure 3e). Enhancers normally closed after the pre-pro B cell stage (note E8, 14 and 15) open in *Brwd1*^{-/-} small pre-B cells. In contrast, there is no change in transcription factor expression (Figures 4b-c). As an example of the impact of these changes, there is greatly diminished binding of IRF4 to enhancers important for late B cell development (Figures 4d-g), even though IRF4 expression is normal both at RNA (Figure 4b) and protein levels (Figure 4c). Our findings clearly demonstrate that BRWD1 has specific effects on the mechanisms of B lymphopoiesis that are fully captured by our experiments.

The editor also requested additional experiments to understand why the function of BRWD1 was restricted to B cells. This very important question was fully addressed in

our previous paper (Mandal, et al., *Nature Immunology*, 2015). The expression of BRWD1 is restricted to the B cell lineage and is repressed by IL-7R-mediated activation. Upon escape from IL-7R signaling in small pre-B cells, *Brwd1* is transcribed and translated. However, this is not sufficient for BRWD1-mediated chromatin binding. Pre-BCR expression, and the activation of ERK, induces the epigenetic landscape required for BRWD1 recruitment to specific chromatin sites marked with H3K9AcS10pK14Ac. In this way, the function of BRWD1 is both lineage and development stage restricted. We regret these points were not clear in the previous manuscript. To address this deficiency, we have made edits to both the introduction (page 4) and discussion (page 15) (both underlined).

REVIEWERS' COMMENTS:

Reviewer #3 (Remarks to the Author):

The authors have been highly responsive to my comments. Their rebuttal letter is well crafted, comprehensive and exhaustive. The edits made to the manuscripts are all to the point and informative. The ms is ready for prime time.